# Integrated Bayesian and Particle Swarm Approaches for Enhanced Gas Leak Detection in Complex Commercial Structures

**DOI:** 10.3390/s25051481

**Published:** 2025-02-28

**Authors:** Zhewen Sui, Xiaobing Yuan, Baoping Cai, Fangqi Ye, Qingqing Duan, Zhiqiang Zhao, Xiaoyan Shao, Xin Zhou, Zhiming Hu

**Affiliations:** 1Shenzhen Urban Public Safety Technology Institute, Shenzhen 518001, China; s23040038@s.upc.edu.cn (Z.S.); 2004011117@s.upc.edu.cn (F.Y.); 2College of Mechanical and Electronic Engineering, China University of Petroleum, Qingdao 266580, China; caibaoping@upc.edu.cn (B.C.); shaoxiaoyan@s.upc.edu.cn (X.S.); zhouxin@s.upc.edu.cn (X.Z.); huzhiming@s.upc.edu.cn (Z.H.); 3Bokang Intelligent Information Technology Co., Ltd., Beijing 100020, China; duanqingqing@enn.cn (Q.D.); zhaozqf@enn.cn (Z.Z.)

**Keywords:** sensor layout optimization, gas leak detection, Bayesian networks, particle swarm optimization

## Abstract

During gas leak detection and risk monitoring of commercial sealed areas, different types of sensors are deployed to monitor leak signals. The arrangement of a limited number of sensors in the most strategic positions, solving the problem of optimal sensor placement, is key to improving detection efficiency. Aiming at gas leak detection in commercial areas, this paper proposes a sensor placement methodology based on a particle swarm optimization algorithm to determine the optimal number and position of sensors. First, Bayesian networks assess the gas leak risk levels. Second, a discrete optimization model for sensor placement is established. Finally, the particle swarm optimization algorithm is applied to calculate the optimal sensor placement solution. In the iterative process, partial differential equation models simulate gas diffusion paths to verify the effectiveness of the sensor layout, followed by computational fluid dynamics simulations for further validation of the optimization results. The simulation case of a commercial gas system demonstrates that the proposed method achieves fast convergence and significant optimization results. A real case study shows that the method reduces the number of sensors and data redundancy. Compared with traditional methods, the robustness and efficiency of the system under the optimal solution are significantly improved.

## 1. Introduction

In commercial sealed areas, gas leaks can lead to serious safety incidents such as fires, explosions, and poisoning [1]. These areas typically feature complex structures and limited ventilation, causing gas to rapidly accumulate and reach dangerous concentrations once a leak occurs, thus increasing the risk of accidents [2,3]. Therefore, timely and accurate detection of gas leaks to ensure that leaks are detected as quickly as possible is critical for the safe operation of these areas.

Traditional gas leak detection systems usually rely on experience or simple coverage calculations for sensor placement. This approach can easily overlook the risk differences in complex environments, leading to inadequate sensor coverage in some high-risk areas and failing to provide effective early warnings at crucial moments. Additionally, traditional methods struggle to optimize sensor layouts in commercial sealed areas with complex architectural structures while balancing detection efficiency and cost control. Therefore, there is a need to develop a sensor placement optimization method that can comprehensively consider various risk factors within the area, improving detection accuracy and efficiency [4].

In the research on sensor layout optimization, numerous studies have explored optimization methods in different application scenarios, achieving significant progress across various fields. For example, the study by Pulthasthan and Pota emphasized that optimizing sensor layout in active noise control systems can significantly improve feedback control efficiency [5]. Azarbayejani et al. [6] proposed a probabilistic method to determine the optimal sensor allocation in structural health monitoring, noting that as sensor technology advances, reasonable placement can enhance the reliability of damage detection. Gao et al. [7] addressed the sensor layout issue for moving vehicle loads on long-span cable-stayed bridges using a hybrid optimization algorithm that combines modal strain energy and adaptive genetic algorithms, improving the accuracy of data collection and signal-to-noise ratio. Liu et al. [8] studied bladder volume measurement based on electrical impedance tomography sensors, optimizing the sensor layout to improve measurement accuracy and reliability. Lin and Chiu [9] proposed a near-optimal sensor layout algorithm based on simulated annealing, achieving full coverage and distinction in sensor networks, demonstrating good practicality, especially in resource-limited scenarios. One study proposed a sensor layout design method based on information entropy, using expected K-L divergence as a measure to optimize sensor placement and improve the accuracy of modal recognition, though it may face challenges related to insufficient information in complex environments [10]. Hu et al. [11] developed an innovative sensor layout method for damage identification in complex structures with structural variations, demonstrating that optimizing sensor placement can significantly enhance the detection capability of structural damage. However, further validation of its feasibility and adaptability in practical applications is needed, particularly for different types of structures and damage modes. Additionally, Chu et al. [12] proposed a probabilistic model based on line-of-sight coverage, considering the directionality and tilt angle of sensors, using a new coverage function to evaluate the effectiveness of sensor layout. This provides new insights into addressing sensor placement issues in complex environments, though the model may have limitations when dealing with dynamic environmental changes. Xiao et al. [13] proposed a sensor layout method based on multiple optimization strategies, emphasizing the importance of reasonable sensor placement in structural health monitoring, showing that proper placement can significantly improve system performance, though balancing sensor quantity, layout cost, and monitoring accuracy remains a challenge in practical applications. Another study explored sensor layout optimization using Bayesian networks, proposing a method based on effective independence to determine sensor positions, maximizing the distance between diagnostic vectors by combining Bayesian inference and optimization algorithms, offering a new perspective on sensor placement [14].

Bayesian networks are statistical inference methods based on probabilistic graphical models that can effectively handle uncertainty and complexity [15]. By constructing Bayesian networks, researchers can assess risks in different areas and generate corresponding risk values for each small area. These risk values not only reflect the potential leakage risks but also provide a scientific basis for sensor placement. The advantage of Bayesian networks is their ability to combine prior knowledge with observational data for inference, especially excelling in handling complex systems [9,10]. For example, research by Yu et al. has shown that Bayesian networks can effectively integrate data from different sensors and consider environmental factors such as temperature, humidity, and wind speed, thereby improving the accuracy of risk assessments [16]. However, Bayesian networks have a high computational complexity, and building the networks requires a large amount of prior knowledge, which may be difficult to obtain in some cases [17,18]. To address these challenges, researchers have proposed various improvements, such as using hybrid models and adaptive algorithms to optimize the construction process of Bayesian networks, thereby enhancing their practical effectiveness [19,20].

Particle swarm optimization (PSO) is a simple, effective, and computationally efficient algorithm inspired by the foraging behavior of birds, making it suitable for real-time optimization in dynamic environments [21,22]. PSO finds near-optimal solutions by leveraging individual and collective experiences, with each particle updating its position in the search space based on shared information. Despite its global search ability, PSO can sometimes fall into local optima, particularly in high-dimensional search spaces, where its performance may be influenced by the initial particle positions [22,23,24]. Moreover, PSO is sensitive to parameter settings, which can significantly impact its effectiveness. To improve performance, researchers have proposed solutions such as diversity-guided algorithms to avoid premature convergence, and large-scale bi-level PSO to enhance global search capabilities [25,26,27,28,29].

Combining Bayesian network risk assessment with PSO can fully leverage the strengths of both. First, Bayesian networks assess risks in the monitoring area, generating risk values for each small area. Then, these risk values are used to dynamically adjust the optimization weights in PSO to optimize sensor placement positions. This method not only improves the efficiency of sensor monitoring but also achieves flexible adaptability under different environmental conditions [30,31,32,33]. Previous studies have shown that PSO exhibits significant advantages in sensor placement optimization. For example, Senouci et al. [34] proposed a method for deploying static wireless sensor networks based on an improved binary PSO algorithm, significantly improving the network’s coverage and energy efficiency. Additionally, Li et al. proposed an optimal deployment algorithm for wireless sensor nodes based on adaptive binary PSO, effectively reducing the number of active nodes, improving coverage uniformity, and enhancing network energy efficiency [35].

Recent advances in electronic nose (e-Nose) technology, particularly through the integration of artificial intelligence (AI) and multi-sensor arrays, have significantly enhanced the detection of combustible gases. The GasCon-Enose system, developed by Attallah et al. [36], uses a combination of feature extraction techniques and hybrid feature selection to achieve high accuracy in gas type and concentration identification. Zhai et al. [37] reviewed the progress in e-Nose technology, highlighting the use of both traditional MOS materials and newer MOFs, along with the importance of addressing sensor drift and improving pattern recognition. Lashkov et al. [38] proposed a low-cost TiO2 nanotube sensor with improved stability, though selectivity remains a challenge. For commercial confined spaces, e-Nose systems can focus on detecting methane, as it is the only combustible gas present. These studies demonstrate the effectiveness of multi-sensor arrays and AI in enhancing gas detection accuracy and reliability [39].

To enhance the monitoring efficiency and safety of the gas leak detection system in commercial sealed areas, this paper proposes an optimized sensor placement method combining Bayesian networks, greedy algorithms, particle swarm optimization algorithms, and partial differential equations (PDEs). To the best of our knowledge, this work systematically applies these methods for the first time to the problem of gas leak sensor placement optimization in commercial sealed areas. The main contributions are summarized as follows:The integration of Bayesian networks and particle swarm optimization allows for gas leak risk assessment and the generation of risk levels for different areas, providing data support for the initial sensor layout. The particle swarm optimization algorithm is then applied to further optimize the sensor layout, ensuring effective coverage and rapid detection in high-risk areas, thus achieving precise and efficient optimization of the gas leak detection system.The sensor layout optimization method combines greedy algorithms and particle swarm optimization algorithms, integrated with PDE models for validation. Initially, the greedy algorithm rapidly generates a preliminary layout, and the particle swarm optimization algorithm ensures effective sensor distribution in high-risk areas. The PDE model simulates the gas diffusion process to validate and adjust the optimization results, significantly improving layout precision and detection efficiency.Computational fluid dynamics (CFD) simulations validate sensor layout effectiveness by accurately replicating gas diffusion in complex environments and optimizing sensor placement based on dynamic leaks. This method enhances layout precision, reliability, and quick response, ensuring efficient monitoring and safety management.

The rest of this paper is organized as follows: Section 2 provides a detailed introduction to the sensor layout optimization method combining Bayesian networks and particle swarm optimization algorithms, incorporating the greedy algorithm and PDE model for optimization validation. Section 3 analyzes specific cases in commercial sealed areas, demonstrating the application effects of the proposed method in sensor placement. Section 4 validates the effectiveness of the optimized sensor layout in actual gas leak scenarios based on simulations and real-world environments. Section 5 summarizes and provides an outlook on the research work presented in this paper.

## 2. Gas Leak Detection Sensor Layout Method

In this study, a method was developed to optimize the layout of gas leak sensors in commercial sealed areas, as illustrated in Figure 1. Initially, the 2D blueprints of the commercial sealed area were divided into regions, each equipped with a Bayesian network for gas leak risk assessment, calculating risk values for each area. Based on these risk values, a sensor layout optimization model was constructed. This model integrated greedy and particle swarm optimization algorithms to adjust the distribution and quantity of sensors, ensuring each area received appropriate monitoring coverage according to its risk level.

For the sensor selection, the authors employed catalytic combustion sensors (model: MQ-5, Hanwei Electronics Co., Ltd., ShenZhen, China), which are sensitive and capable of detecting methane concentrations as low as 300 ppm (20% of the lower explosive limit, LEL). The sensor’s working principle is based on the flameless combustion of methane, generating a voltage signal proportional to the gas concentration. A binary alarm is triggered when the methane concentration reaches 20% LEL, providing a rapid response within 3 s. These sensors were calibrated according to manufacturer specifications, and their range was optimized based on gas diffusion time simulations via PDE and CFD models. However, in the context of commercial closed spaces, where the only combustible gas leakage source is natural gas, which primarily consists of methane, there is no need for additional gas selection considerations. The primary focus of the detection system is to monitor methane, as it is the main component of natural gas.

Additionally, the optimization process incorporated PDE to simulate gas diffusion, validating the effectiveness of each generation of optimization results. The constraints of the optimization included the minimum distance between sensors, the maximum coverage area, and the response time. This method effectively determined the optimal layout of sensors.

Finally, CFD technology was used to validate the sensor positions in each area, ensuring they could detect gas leaks quickly and accurately within critical windows of opportunity, thus enhancing the safety and efficiency of the system.

### 2.1. Application of Bayesian Networks in Gas Leak Risk Assessment

In this study, a Bayesian network model was constructed to assess the risk of gas leaks, incorporating factors from multiple levels such as equipment status, environmental conditions, and operational standards. Top-level nodes like gas leaks and equipment status are divided based on different stages and potential impact factors, while child nodes such as pipeline status, valve condition, and operator experience are determined by their conditional probabilities related to parent nodes. This model integrates structural and parametric modeling, particularly using fuzzy set theory to determine the prior probabilities of root nodes in the network in the absence of historical data, ensuring the accuracy of the analysis. Through this approach, potential risks of gas leaks can be systematically identified and quantified, providing a scientific basis for preventive measures and emergency responses, effectively enhancing system safety and response efficiency.

#### 2.1.1. Determining the Fuzzy Probabilities of Root Nodes

The fuzzy set theory is an extension of traditional set theory methods used to handle the fuzziness and subjectivity of human judgments. This theory employs fuzzy numbers, such as triangular fuzzy numbers, trapezoidal fuzzy numbers, Gaussian fuzzy numbers, and LR-type fuzzy numbers, to express imprecise values. Triangular fuzzy numbers are commonly used to determine the fuzzy probabilities of root nodes due to their ease of reference function processing and simplified algebraic operations. The membership function of fuzzy numbers describes their uncertainty, with values ranging from 0 to 1.

To ensure the objectivity of the assessment, this study gathered a group of domain experts with diverse professional backgrounds, lengths of experience, educational levels, and ages. The role of these experts was crucial in helping define the fuzzy probabilities for the root nodes based on their deep understanding of the domain. Their assessments were used to fine-tune the values of the root nodes in the Bayesian network. To reduce the subjective bias of expert judgments, the weights *r_j_* of each expert were calculated based on the data in Table 1 using the following quantification method:(1)$$r_{j}=\frac{\theta_{j}}{\sum_{j=1}^{n}\theta_{j}}$$
where *r_j_* represents the weight of the *j*-th expert; *θ_j_* is the total score of that expert, which is calculated by summing the scores for the expert’s professional position, years of experience, educational level, and age.

The experts’ opinions are aggregated into a triangular fuzzy number Wi using the linear opinion pool method:(2)$$W_{i}=\sum_{i=1}^{m}r_{j}A_{ij}j=12\ldots n$$
where *W_i_* is the aggregated triangular fuzzy number for event *i*; *r_j_* represents the weight of the *j*-th expert; *A_ij_* is the triangular fuzzy number assessment of event *i* by that expert; *m* is the total number of events, and *n* is the total number of experts involved in the assessment. This expression is more concise and clearer, highlighting the meaning of each variable and their role in the assessment process.

Using the linear opinion pool method detailed in Table 2, the experts’ opinions are aggregated into a triangular fuzzy number W_i_:(3)$$W_{i}=\sum_{i=1}^{m}r_{j}A_{ij}j=12\ldots n$$
where *W_i_* is the aggregated triangular fuzzy number for event *i*; *r_j_* represents the weight of the *j*-th expert; *A_ij_* is the triangular fuzzy number assessment for event *i* by that expert; *m* is the total number of events, and *n* is the total number of experts involved in the assessment. This statement clearly elucidates the roles and importance of each variable in the data processing.

The calculations result in an interval. To quantify this interval, the centroid method is used to determine the fuzzy possibility (FPs) of the root nodes. The calculation method is as follows:(4)$$FPs=\frac{\int \mu_{i}(x)xdx}{\int \mu_{i}(x)dx}$$
where *μ*_i_(*x*) is the membership function of the triangular fuzzy number, represented as:(5)$$\mu x=\left\{\begin{array}{c} \begin{array}{cc} 0 & x<a \end{array}\\ \begin{array}{cc} \frac{x-a}{m-a} & a\leq x\leq m \end{array}\\ \begin{array}{cc} \frac{b-x}{b-m} & m\leq x\leq b \end{array}\\ \begin{array}{cc} 0, & x>b \end{array} \end{array}\right.$$
where *a*, *m*, and *b*, respectively, represent the minimum, median, and maximum values of *x*.

Finally, the fuzzy possibilities (FPs) need to be converted into fuzzy probabilities (FPr), which serve as the prior probabilities for the root nodes in the Bayesian network:(6)$$FPr=\frac{1}{10^{k}}k={\left(\frac{1-FPs}{FPs}\right)}^{\frac{1}{3}}\times 2.301$$

#### 2.1.2. Determining the Conditional Probability Table for Child Nodes

The conditional probability table (CPT) is used to express the dependency relationship between parent nodes and child nodes. For child nodes involving equipment factors, environmental factors, and third-party interference, if any of the parent nodes fail, the corresponding child node will also fail. This relationship can be described using OR logic. However, when human and management factors are involved, the relationship between child nodes and parent nodes is not a simple logical relationship but needs to be represented by truth values. In such cases, the noisy-OR model can be used to calculate the conditional probabilities. The calculation formula is:(7)$$P\left(T|X1X2\ldots Xn\right)=1-\prod_{1\leq j\leq n}\left(1-P_{j}\right)$$

#### 2.1.3. Posterior Probability and Mutual Information Determination

Using the causal inference capabilities of Bayesian networks, the probability of top-node events occurring under the influence of multiple risk factors can be calculated. This predictive analysis method relies on the structure of the Bayesian network, which models the causal relationships between different factors. This way, it is possible to predict the probability of events occurring under the combined effects of various risk factors.

Additionally, using Bayes’ theorem for diagnostic analysis, the posterior probabilities of other related events can be calculated when one or more specific events are known to have occurred. This method helps determine the likelihood of each event under specific conditions.

Finally, through mutual information methods and sensitivity analysis, the impact of various factors on the event outcomes can be determined. This analysis helps reveal which factors have the most significant impact on the event, thereby providing a scientific basis for risk management and decision making.

The formula for calculating marginalized probabilities is as follows:(8)$$P\left(X_{i}\right)=\sum_{exceptX_{i}}P\left(U\right)P\left(U\right)=\prod_{i=1}^{n}P\left(X_{i}|Pa\left(X_{i}\right)\right)$$
where *Pa*(*X_i_*) represents the set of parent nodes for the variable *X_i_*.

Given a set of observed values for variables *E*, considered as evidence, the posterior probability (PT) for a specific variable can be calculated based on Bayes’ theorem. Bayes’ theorem provides a method for updating the probability estimates of events occurring, given some known conditions. The mathematical expression for Bayes’ theorem is:(9)$$P\left(U|E\right)=\frac{P(E|U)P(U)}{P(E)}=\frac{P\left(EU\right)}{\sum_{U}P\left(EU\right)}$$

Mutual information (MT) is a measure of the degree of dependency between two random variables and is also used to quantify the reduction in uncertainty of one random variable given the information about another. Therefore, it is commonly used to determine the importance of events or variables. The formula for calculating mutual information is as follows:(10)$$I\left(T,X\right)\sum_{x}\sum_{t}P\left(tx\right)\log\frac{P\left(tx\right)}{P\left(t\right)P\left(x\right)}$$
where *P*(*t*, *x*) is the joint probability distribution function of *t* and *x*, and *P*(*t*) and *P*(*x*) are the marginal probability distribution functions of *t* and *x*, respectively.

### 2.2. Sensor Layout Optimization Based on Greedy Particle Swarm Algorithm

In the process of optimizing sensor placement, the greedy algorithm is initially used for preliminary optimization. The greedy algorithm selects the best current option at each step, quickly forming an initial solution that provides a reasonable starting point for the basic layout of the sensors. Subsequently, the particle swarm optimization algorithm is employed to further refine and improve upon these initial results. The particle swarm algorithm utilizes principles of swarm intelligence, updating the positions and velocities of particles through iterations to find superior solutions. This strategy, which combines the greedy algorithm and particle swarm algorithm, ensures that the entire optimization process is efficient and progressively approaches the global optimum.

#### 2.2.1. Preliminary Optimization with Greedy Algorithm

In the optimization process of this study, the greedy algorithm is initially used as the first step in layout optimization. Specifically, the operation involves randomly selecting n points on the map, and choosing one of these points as the initial position P for a sensor. The algorithm then iterates over these points, calculates their distance from the sensor position P, and adds the furthest point to the sensor layout list. This process is repeated until the predetermined coverage rate is achieved. Through this method, a preliminary sensor placement scheme can be quickly obtained. Although this scheme does not fully consider the effects of wall diffusion, it is sufficient to serve as the starting condition for the particle swarm algorithm, reducing the randomness in the generation of initial conditions for the particle swarm algorithm. The rough coverage range of the sensor is:(11)G=S×T
where *S* represents the operating speed of the gas, and *T* denotes the time it takes for the sensor to receive the gas alarm.

#### 2.2.2. Particle Swarm Algorithm for Layout Optimization

As shown in Figure 2, the PSO algorithm searches for the optimal solution by moving particles (representing potential sensor positions) within the search space, continuously updating their positions based on both their own experiences and the experiences of other particles. The advantage of PSO lies in its ability to extensively search for possible sensor layouts while effectively converging to the optimal solution. It balances the need for a comprehensive global search with the fine-tuning of sensor layouts in high-risk areas. By iterating through multiple possible solutions and adjusting the positions and velocities of the particles, PSO ensures that the final sensor layout maximizes coverage while minimizing the number of sensors used.

The core of the optimization lies in adjusting the fitness function, namely the sensor coverage function, to achieve multi-objective optimization, i.e., maintaining a minimum distance of over 8 m between each sensor while minimizing the number of sensors used and maximizing area coverage. The particle swarm optimization function is:(12)$$v_{i}^{\left(t+1\right)}=w\cdot v_{i}^{\left(t\right)}+c_{1}\cdot r_{1}\cdot (p_{best,i}-x_{i}^{\left(t\right)})+c_{2}\cdot r_{2}(g_{best}-x_{i}^{\left(t\right)})$$
where $v_{i}^{\left(t\right)}$ is the velocity of particle *i* at time *t*. *w* is the inertia weight, controlling the magnitude of velocity changes. *c*_1_ and *c*_2_ are learning factors. *r*_1_ and *r*_2_ are random numbers within the range [0, 1]. *P_best_* is the best position found so far by particle *i*. *g_best_* is the best position found so far by the entire swarm.

In this study, the PSO algorithm plays a crucial role in optimizing sensor layout. To ensure that PSO can effectively search for the optimal solution, its parameters must be carefully set and adjusted. These parameters include the velocity limit Vmax, inertia weight w, cognitive learning factor *c*_1_, and social learning factor *c*_2_. These elements significantly influence the algorithm’s global and local search capabilities.

Firstly, to prevent particles from deviating from the feasible search space during the search process, the velocity V of particles in each dimension is typically constrained within the range [−Vmax, Vmax], Vmax. The choice of the upper velocity limit Vmax directly affects the global and local search capabilities of PSO. If Vmax is set too high, particles may fly out of the solution space, leading to an ineffective convergence to the optimal solution; conversely, if Vmax is too low, the movement of particles might be insufficient, potentially trapping them in local optima and preventing escape. Therefore, Vmax is usually set as a proportion of the range of variation of the variables in each dimension, to balance between global and local search capabilities.

Secondly, the inertia weight w is a critical parameter that controls the magnitude of velocity changes during the update process. The function of inertia weight is to gradually decrease the particle’s momentum throughout the iterations, thus shifting the search from global exploration to local exploitation. In this study, the initial value of the inertia weight w is set to a relatively high value to promote global search; as the iterations progress, w is gradually reduced to a lower value to enhance the local search capabilities, ensuring that the algorithm can accurately find the optimal solution during the convergence process.

The cognitive learning factor *c*_1_ and the social learning factor *c*_2_ are crucial parameters in PSO that influence the direction of particle updates. The cognitive learning factor *c*_1_ controls the extent to which particles move towards their own historical best positions, while the social learning factor *c*_2_ governs the extent to which particles move towards the swarm’s best position. By adjusting the values of these two parameters, the balance between individual exploration and collective information sharing can be managed. In this study, both *c*_1_ and *c*_2_ are set to relatively high values to ensure that particles can fully utilize individual experience and collective information during each update, thereby accelerating the convergence speed.

During the specific optimization process, the positions and velocities of particles are updated in each iteration based on the aforementioned parameters. However, to prevent the particles from exceeding the boundaries of the search space, if a particle exceeds the maximum position limit Xlimit during the update process, a correction algorithm is required to adjust the particle’s position to keep it within the feasible search range. This correction mechanism helps maintain the effectiveness of the particles within the search space, preventing the algorithm from failing or falling into invalid solution areas.

In the early stages of particle swarm optimization, selecting an appropriate number of particles and iterations can effectively explore the solution space and avoid falling into local optima. As the algorithm iterates, the gradual adjustment of the inertia weight shifts the focus of the search towards the vicinity of the global optimum, thereby achieving the optimal sensor layout. Through such fine-tuning of parameters, the particle swarm optimization algorithm ensures both the breadth of global search and the precision of local search.

From Figure 3, it is evident that in the discrete particle swarm optimization process, the maximum inertia weight *w*_max_ is set to 0.9, and the minimum inertia weight *w*_min_ is set to 0.4. Both the cognitive learning factor *c*_1_ and the social learning factor *c*_2_ are set to 2.0. The maximum number of iterations for the algorithm is set to 100, and the population size N is 20. For particles that exceed the maximum position limit Xlimit, a correction algorithm is employed to adjust their positions.

Then, the position of each particle is updated according to the following formula:(13)$$x_{i}^{\left(t+1\right)}=x_{i}^{\left(t\right)}+v_{i}^{\left(t+1\right)}$$

### 2.3. PDE Simulation to Refine Optimization Results

In this study, to comprehensively consider the impact of buildings on gas diffusion in the actual environment, a plume dispersion model was specifically used to simulate the gas diffusion at locations prone to leaks, such as bends and flanges in gas pipelines. Using the PDE toolbox in MATLAB2021a, a detailed two-dimensional partial differential equation (PDE) model was successfully constructed. The gas diffusion equation is as follows:(14)$$C(x,y,z)=\frac{Q}{\left(2\pi \sigma_{y}\sigma_{z}u\right)}\exp(-\frac{y^{2}}{2\sigma_{y}^{2}})\exp(-\frac{{\left(z-H\right)}^{2}}{2\sigma_{z}^{2}})$$
where *C*(*x*, *y*, *z*) represents the concentration of pollutants at point (*x*, *y*, *z*); *Q* is the emission rate of the pollutant source; *σ_y_* and *σ_z_* are the standard deviations in the horizontal and vertical directions of the Gaussian distribution, representing the spread of the pollutant; *u* is the wind speed; *H* is the height of the pollutant source; *x*, *y*, and *z* are the coordinates in space, where *x* is the downwind distance, *y* is the lateral distance, and *z* is the vertical distance.

This model creates complex geometries by defining a geometric description matrix (GDM) and applying the decsg function. Furthermore, coefficients for the PDE equation are set, and the equation is solved by generating a mesh. The key role of this model is to verify whether the selected sensor layout can detect a specific concentration of gas within a given time frame. If sensors at certain locations fail to meet this requirement, the algorithm will re-optimize until the most suitable sensor layout is found. By combining physical models with the algorithmic optimization process, this method ensures that the sensor layout is not only theoretically optimal but also effective in practical scenarios.

### 2.4. CFD Simulation to Validate Results

The jet velocity can be derived from the isentropic jet velocity equation as follows:(15)$$v=\sqrt{\frac{2k}{k-1}\cdot R\cdot T\cdot \left[1-{\left(\frac{P_{A}}{P}\right)}^{\frac{k-1}{k}}\right]}$$
where *v* is the jet velocity at the leakage point; *k* is the isentropic index of the gas; *R* is the gas constant; *T* is the absolute temperature; *P* is the absolute pressure inside the pipeline; and *P_A_* is the ambient pressure.

The mass flow rate can be calculated using the following equation:(16)$$\dot{m}=\rho \cdot A\cdot v$$
where $\dot{m}$ is the mass flow rate; *ρ* is the gas density; *A* is the area of the leakage point; and *v* is the jet velocity at the leakage point.

The convection–diffusion equation can be expressed as [37]:(17)$$\frac{\partial \varphi }{\partial t}+u\cdot \nabla \varphi =D\nabla^{2}\varphi$$
where *φ* is the gas concentration; *t* is time; *u* is the velocity field of the fluid (in this case, including the jet velocity); *D* is the diffusion coefficient of the gas; ∇*φ* and ∇^2^*φ* are the concentration gradient and the second-order derivative, respectively.

## 3. Sensor Layout Optimization Example for Commercial Sealed Areas

### 3.1. Risk Assessment of Sealed Areas

As the initial stage of algorithm optimization, this study employs image processing techniques to preprocess the floor plan of the commercial area. Using edge detection and image segmentation techniques, key infrastructure such as walls and pipelines are identified, establishing spatial constraints for sensor placement. This step is crucial for the accuracy of the subsequent analysis. Next, the floor plan is divided into a 5 × 5 grid within the MATLAB environment, allowing for a detailed risk assessment of each cell. Based on this, the fuzzy set theory is used for risk assessment, converting uncertain risk factors into specific risk values and constructing a comprehensive risk matrix, which provides a critical foundation for optimizing sensor placement.

The risk values are further refined using expert input, as described in Section 2.1.1. The experts’ knowledge allows for more accurate definition of the fuzzy probabilities for each risk factor, ensuring a more robust and contextually relevant risk assessment. This step ensures that domain-specific knowledge directly informs the Bayesian network’s evaluation of risk in the commercial sealed areas.

A Bayesian network model was developed to assess gas leak risks in the commercial sealed area, as illustrated in Figure 4. This model systematically quantifies and evaluates the multiple factors that influence gas leakage, revealing their contribution to gas leak risks through the dependencies among these factors, as shown in Table 3 and Table 4.

The expert-driven refinement of the fuzzy probabilities enables the Bayesian network to produce more accurate and dynamic risk assessments, thus guiding the optimization of sensor placements by accounting for complex real-world variables.

First, the top-level node in the Bayesian network, “Gas Leak” (M1), is identified and defined. This is the core event being evaluated, with all other factors constructed around this node. Next, four key aspects that directly influence gas leakage are determined: equipment condition (M2), environmental conditions (M3), operational procedures (M4), and maintenance activities (M5). Each of these major aspects is further subdivided into multiple sub-nodes to describe more specific influencing factors.

For the equipment condition (M2), it is further refined into pipeline condition (X1), pipeline material (X2), pipeline joints (X3), and pipeline pressure (X4). These sub-nodes primarily reflect the physical state of the pipeline system itself. If issues arise in any of these areas, the risk of gas leakage increases significantly.

Environmental conditions (M3) encompass factors related to the detection equipment and the surrounding environment, such as valve condition (X5), sensor status (X6), sensor type (X7), sensor location (X8), temperature conditions (X9), humidity conditions (X10), ventilation status (X11), ventilation system (X12), and space volume (X13). These nodes describe the external environment and the functional state of the equipment, directly affecting the efficiency of gas leak detection and the speed and extent of gas diffusion. For example, temperature and humidity influence the speed of gas diffusion in the air, while sensor location and type determine the efficiency of detecting leaked gas.

Under the node for operational procedures (M4), the sub-nodes primarily involve human operations and management, such as operator experience (X14), maintenance frequency (X15), type of maintenance (X16), frequency of external inspections (X17), and compliance with operational procedures (X18). These nodes capture the uncertainties related to operational aspects, including the skill levels of operators, the frequency and quality of maintenance activities, and the strict adherence to operational procedures. Variations in these factors will directly affect the probability of gas leaks occurring.

In terms of maintenance activities (M5), factors considered include emergency response plans (X19), training frequency (X20), types of maintenance (X21), and frequency of external inspections (X22). These describe the adequacy and effectiveness of preventative and emergency measures before and after a gas leak occurs. Frequent and high-quality maintenance and training can significantly reduce the risk of gas leaks, and a well-prepared emergency plan can provide a rapid response and mitigate the consequences of a leak.

The construction of the entire Bayesian network is a process that unfolds layer by layer from the top-level event (gas leak) downward. After identifying the top-level node and its direct influencing factors, the various main influencing factors are further refined to form a complete causal chain. The selection of sub-nodes during this process is based on a deep understanding of the gas leak process and the opinions of experts in related fields, ensuring the comprehensiveness and accuracy of the model.

Through this Bayesian network model, it is possible not only to quantitatively assess the contribution of various factors to the risk of gas leaks but also to predict the probability of a gas leak occurring under certain known conditions (such as an increase in environmental temperature or abnormal pipeline pressure). This structured risk assessment tool provides strong support for decision makers, helping them to identify potential risks early and take preventative measures to reduce the occurrence and harm of gas leak incidents.

In the probability tables of the Bayesian network model detailed in Table 4, the impact of each node on the risk of gas leaks is quantified through prior probabilities (FPrs), posterior probabilities (PTs), and mutual information (MT). For instance, at node X2 (pipeline material), the prior probability for PVC is 0.2%, the posterior probability is 5.5%, and the mutual information value is 0.5, indicating a significant influence in risk assessment. Similarly, other nodes like X3 (pipeline joint condition) and X4 (pipeline pressure) also display probability variations under different conditions in the table, providing crucial data for assessing and responding to the risk of gas leaks.

Based on the calculation results for various conditions in the table, 1 area out of 25 is selected for comprehensive consideration of factors, including steel pipeline material, aging pipeline joints, low pipeline pressure, corroded valve conditions, normal sensor status, normal temperature conditions, normal humidity conditions, poor ventilation, experienced operators, regular maintenance frequency, partial compliance with operational procedures, established emergency plans, and regular training frequency. The final calculated probability of a gas leak in this area of the commercial sealed zone is approximately 6.84%. Similarly, the risk probabilities for all 25 sections of the entire commercial sealed area are displayed in Figure 5.

Based on the risk heat map illustrated in Figure 6, the risk value R is divided into four levels, as detailed in Table 5. An initial number of sensors is allocated to each area accordingly. A multi-level decision-making approach is used to provide preliminary guidelines for the quantity and layout of sensors. For example, different sensor coverage levels are assigned to areas with different risk levels, ranging from 0 in low-risk areas to 3 in high-risk areas, ensuring effective allocation of resources. This tiered approach facilitates targeted monitoring where it is most needed, optimizing both safety and efficiency in sensor deployment within the commercial sealed area.

### 3.2. Layout Optimization Using Greedy Particle Swarm Optimization Algorithm

In the process of layout optimization using the greedy particle swarm algorithm, key functions include the coverage count function and the coverage rate function. The coverage count function calculates the coverage based on the target area’s longitudinal and latitudinal coordinates (m and n), while the coverage rate function assesses the overall coverage based on the total number of sensors N and the dimensions of the target area (length L1 and width L2).

The coverage count function is:(18)$$C=\left\{\begin{array}{c} 1,D\leq R,\mathrm{Point}\;(m,n)\;\mathrm{is}\;\mathrm{within}\;\mathrm{the}\;\mathrm{range}.\\ 0,D>R,\mathrm{Point}\;(m,n)\;\mathrm{is}\;\mathrm{out}\;\mathrm{of}\;\mathrm{range}. \end{array}\right.$$
where *m* is the horizontal coordinate of the target area and *n* is the vertical coordinate of the target area.

The coverage rate function is:(19)$$Z=\sum_{i=1}^{N}\frac{\sum_{m=1}^{L}\sum_{n=1}^{L}C(m,n)}{L_{1}*L_{2}}$$
where *N* represents the total number of sensors; *L*_1_ and *L*_2_ represent the lengths of the sides of the target area.

The core of the optimization lies in adjusting the fitness function, which is the sensor coverage rate function, to achieve multi-objective optimization. This includes maintaining a minimum distance of over 8 m between each sensor, while simultaneously minimizing the number of sensors used and maximizing the area coverage rate. This strategic approach ensures an efficient and effective distribution of sensors, optimizing both the resources used and the security provided by the detection system.

The simulation of the gas diffusion process is conducted using the PDE toolbox. The optimization objective is to ensure that sensors can detect a sufficient concentration of gas within a designated time frame after a leak occurs, thereby enabling timely warning signals. In practice, the particle swarm optimization algorithm continuously adjusts the positions of the sensors to optimize the layout. After each layout adjustment, the PDE toolbox is used to simulate the diffusion of gas within the specified area, and the concentration changes detected by each sensor at different locations are recorded. This iterative process helps refine sensor placement to maximize coverage and response effectiveness in detecting gas leaks.

Figure 7 illustrates the simulation test results after sensor position adjustments through detailed optimization iterations in three stages. The objective of each stage is to verify whether the sensors can detect a gas concentration of at least 0.002 within 150 s.

In the first stage, the concentration grows slowly and does not reach the predetermined value, which may be due to the sensors being too far from the leak source or the path of gas flow to the sensors being obstructed. Environmental factors such as unfavorable air flow directions might also affect the detection efficiency of the sensors.

The second stage shows a quicker initial increase but still fails to meet the standard, indicating that while the preliminary position adjustment improves response speed, it is still insufficient to reach the required concentration within the limited time. This could be due to obstacles between the sensors and the leak point.

The success of the third stage demonstrates further optimization of sensor positions and possible improvements in sensor technology, such as increased sensitivity, which rapidly raises the concentration to 0.025, far exceeding the minimum requirement. This achievement underscores the importance of meticulous adjustments and technological upgrades in achieving rapid and effective detection.

Throughout the entire optimization process, if the simulation results indicate that the sensors do not meet the expected concentration requirements within the designated time, the particle swarm algorithm automatically adjusts the sensors’ positions, rearranges the layout, and continues optimization until the monitoring requirements are satisfied.

Through precise algorithm training and validation, the final sensor placement matrix H is obtained. Notably, a coverage rate of 97% is achieved after just 10 iterations, demonstrating the efficiency of this research method. Particularly, comparison graphs between the combined greedy and PSO method (Greedy-PSO) and the sole PSO method show that the combined approach not only accelerates the improvement of coverage rate but also achieves a high coverage rate rapidly in the early iterations, displaying a significant advantage. The Greedy-PSO curve has a steeper slope in the early iterations, indicating a faster growth rate in coverage. In contrast, the PSO algorithm’s curve has a gentler slope, signifying a slower increase in coverage. However, as the number of iterations increases, the coverage rates of both methods converge, highlighting the efficiency advantage of the combined algorithm in the early iterations.

In Figure 8, with the simulated annealing algorithm, the Greedy-PSO method quickly achieves a 97% coverage rate after about 10 iterations, while the coverage growth of the SA algorithm is relatively slow. This rapid growth in the initial phase emphasizes the efficiency of the combined algorithm in the early optimization process and its significant advantage in quickly achieving high coverage rates. Additionally, the floor plan shown in Figure 9 demonstrates the practical application of the algorithm in real spatial layouts, with sensor positions optimized and evenly distributed at key monitoring points to accommodate the effects of building layouts on gas diffusion. This layout optimization not only enhances monitoring efficiency but also underscores the algorithm’s adaptability and effectiveness in handling complex real-world constraints.

## 4. Ansys Simulation for Validating the Accuracy of Sensor Optimization Results

In this study, detailed CFD simulations are conducted for 1 of the 64 subdivided small grid areas to analyze the leakage process within a specific region. This area, due to its complex pipeline configuration that includes bends, flanges, and joints among other potential leakage points, as illustrated in Figure 10, becomes the focal point for the simulation analysis.

Under the simulation assumptions, the pressure of the gas pipelines in the commercial complex is set at 0.2 MPa, assuming that a leak occurs at a bend in the pipe with a leak area of 1 cm^2^. The calculated jet velocity is 326.28 m/s, based on the gas’s isentropic exponent of 1.4, the gas constant of 287 J/(kg·K), room temperature of 300 K, and atmospheric pressure. Furthermore, the total mass flow rate of the leak, determined by the jet velocity and the leak area, is calculated to be approximately 42 g/s. By selecting bends, flanges, and joints as the leak points, the CFD simulation considers the gas diffusion process under the jet velocity conditions, using the convection–diffusion equation to integrate the important processes of convection (flow of gas due to jet velocity) and diffusion (movement of gas molecules due to concentration gradients).

The settings of the CFD simulation shown in Figure 11 reflect realistic scenarios where the leak points are oriented towards the interior, simulating the process of gas leaking into a building. The species transport model used in the simulation accurately analyzes the mixture of methane gas and air. The SIMPLEC solver method, along with the settings for the time step and iteration count, ensures the accuracy and stability of the simulation results.

By deploying sensors in critical areas, methane concentration components are monitored. When the methane concentration detected by the catalytic combustion sensor reaches 20% of the lower explosive limit (LEL), the sensor triggers an alarm. The catalytic combustion sensor’s response is a transition from 0 to 1, with no alarm triggered until the concentration reaches 20% of the LEL, ensuring its effectiveness in detecting leaks that pose a risk of explosion. Figure 12 illustrates the relationship between gas leak concentration and time from three different leak points to a fixed sensor position. The curve for leak point one rises quickly to a peak and then stabilizes, indicating that the path between this point and the sensor is either direct or relatively close, yet it ultimately does not exceed the detection threshold, possibly due to the leak location not being conducive to gas accumulation. The curve for leak point two, although rising more slowly, reaches a higher peak, suggesting a slightly longer or more obstructed path. The curve for leak point three rises the slowest and has the lowest peak, indicating that it is the furthest or the most obstructed among the three points. By analyzing the rise speed and peak concentrations of these curves, we can infer the relative distances and potential obstacles between each leak point and the sensor, which is crucial for optimizing sensor layouts and enhancing gas leak emergency response strategies.

According to the simulation results, the monitoring data under the original sensor arrangement show that not all leak points reach the set threshold concentration, reflecting potential monitoring blind spots or delayed detection responses in some areas. Figure 13 shows the relationship between gas leak concentration and time from the three different leak points after optimizing sensor positions. Leak point one displays a relatively stable rising trend and reaches a concentration of 0.84%, indicating a direct and continuous monitoring path between the sensor and the leak point. The curve for leak point two shows significant fluctuations, rising sharply, then falling, and rising again, possibly due to obstacles or airflow changes between the sensor and the leak point. Leak point three quickly reaches a peak initially and then rapidly declines, suggesting a possible significant distance between the leak source and the sensor or a significant change in leak conditions.

The optimized layout observes that the concentration curves for all leak points rise to or exceed the 0.002 standard within 100 s. This rapid response time, within a set emergency window of 3 min, provides an ample margin for reaction, thus confirming the effectiveness of the sensor layout optimization. Additionally, after reaching the threshold, the concentration curves show a stable or declining trend, indicating improvements in data stability in the optimized system.

The optimized results shown in Figure 14 demonstrate that the adjusted sensor layout can rapidly and accurately detect gas leaks within the critical window period, with detection efficiency improving by up to 47% compared with traditional layouts. Here, detection efficiency is defined as the time required for gas to diffuse from potential leakage points to the sensors and reach the detection threshold. The optimization ensures that the time taken for gas concentration to reach the sensor’s detection threshold (e.g., 20% of the lower explosive limit) is minimized. In comparison, traditional layouts typically place sensors based on installers’ experience, often without considering gas diffusion dynamics, leading to less efficient coverage and slower detection times. In these traditional setups, sensors are typically located in predetermined positions with a limited view of high-risk areas, resulting in longer times for gas to diffuse and trigger alarms.

The algorithmic optimization significantly enhances the monitoring capability of the gas leak detection system by strategically positioning sensors in areas with a higher risk of gas accumulation. This improvement not only boosts the system’s alert efficiency at crucial moments but also ensures more reliable and faster detection of leaks. This progress not only showcases the value of combining computational fluid dynamics (CFD) with algorithmic optimization in practical applications but also lays a solid foundation for further research in this field.

## 5. Conclusions

This study successfully developed a sensor layout optimization method for gas leak detection in commercial sealed areas by integrating Bayesian networks and a greedy particle swarm optimization algorithm. By constructing a Bayesian network model, this study systematically assessed the impact of various factors such as equipment condition, environmental conditions, and operational standards on the risk of gas leaks. Through the combination of the greedy algorithm and particle swarm optimization, the sensor layout was precisely optimized, achieving efficient risk monitoring and rapid leak detection. The optimized results demonstrated that the new sensor layout improved detection efficiency by up to 47% during the critical window period compared with traditional layouts, significantly enhancing system safety and response speed. Additionally, the effectiveness of the optimized sensor layout was validated through PDE simulation, ensuring rapid and accurate gas leak detection in practical applications. This research not only improved the efficiency and accuracy of gas leak detection but also provided a scientific decision support tool for safety management in commercial sealed areas, contributing new perspectives and methods to the field’s research and practice.

Future work will focus on further reducing system costs, enhancing the universality and robustness of the algorithm, and exploring the application of this method in other similar commercial environments. Considering environmental protection and sustainable development, research will also be conducted on environmentally friendly sensor materials and their application in the system.

## Figures and Tables

**Figure 1 sensors-25-01481-f001:**
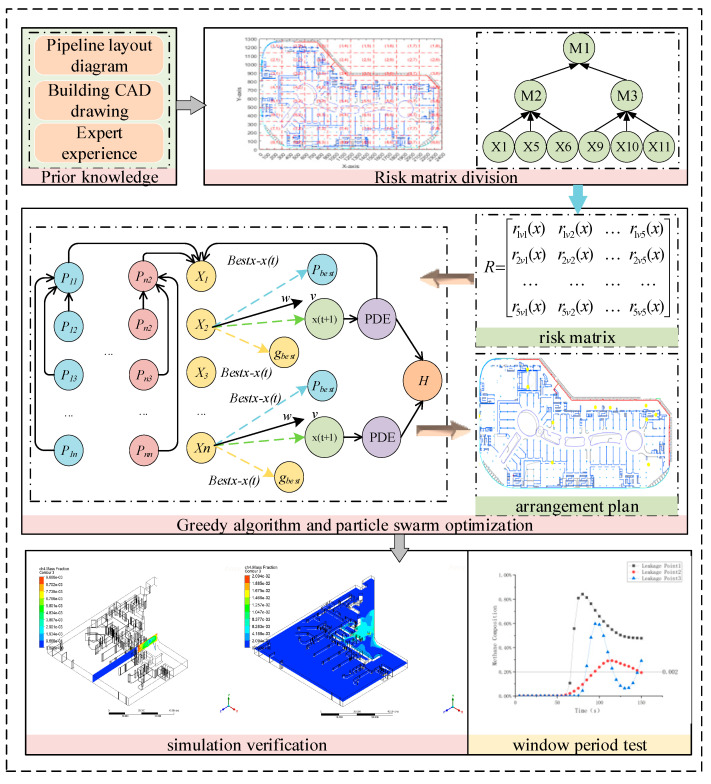
Optimization process method diagram.

**Figure 2 sensors-25-01481-f002:**
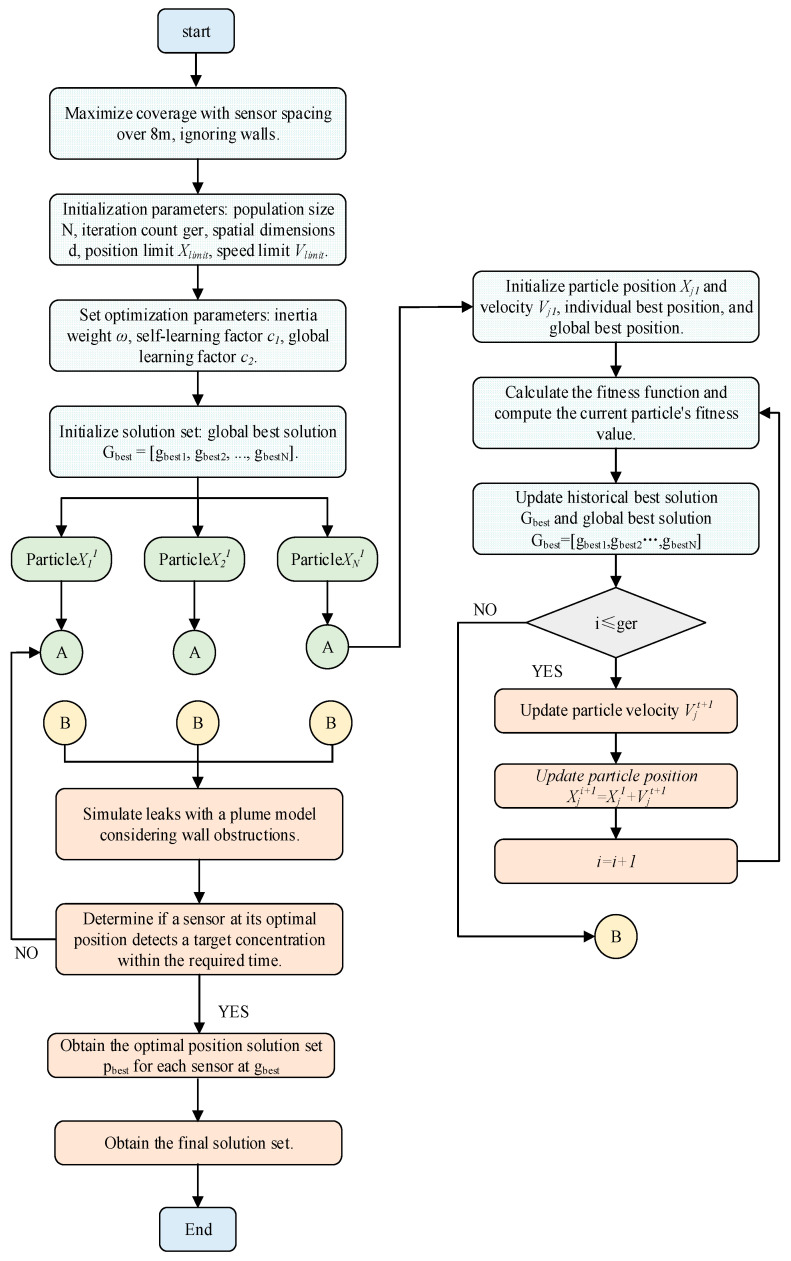
PSO flowchart under the influence of gas diffusion.

**Figure 3 sensors-25-01481-f003:**
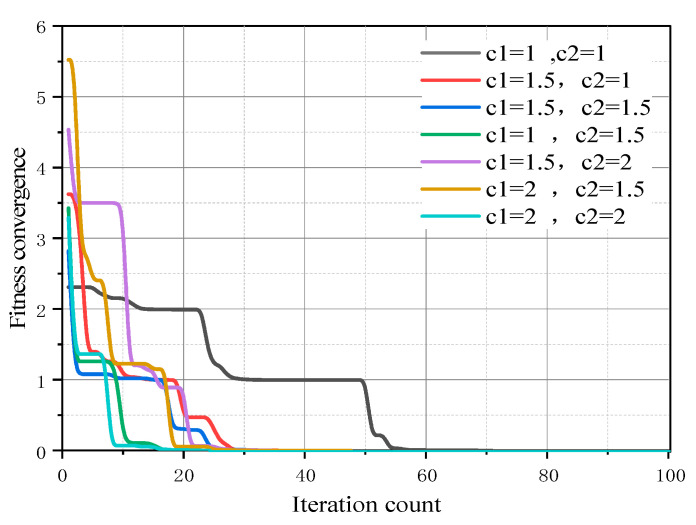
Convergence graph of particle swarm algorithm.

**Figure 4 sensors-25-01481-f004:**
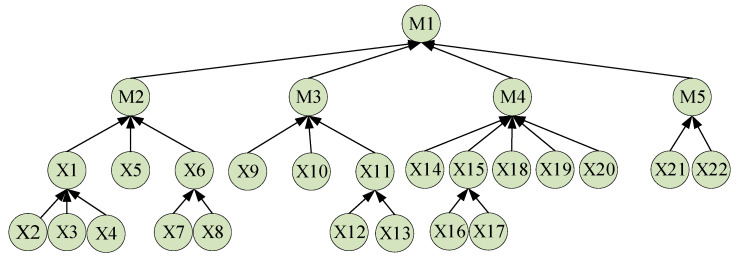
Bayesian network model for gas leak in commercial area.

**Figure 5 sensors-25-01481-f005:**
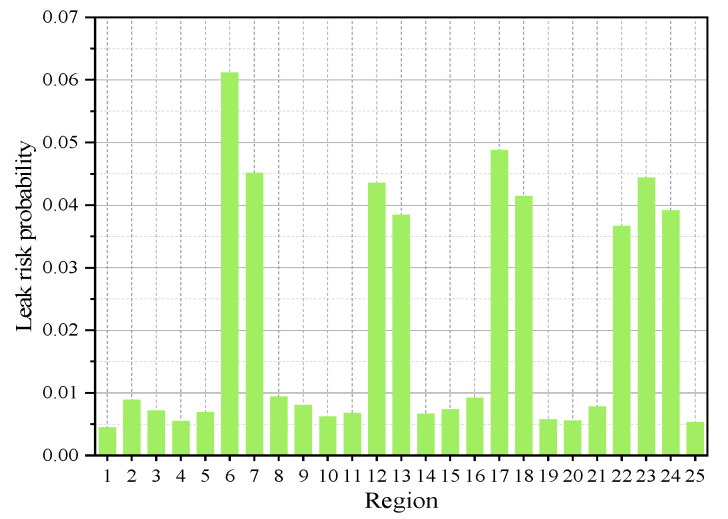
Risk probability of each area.

**Figure 6 sensors-25-01481-f006:**
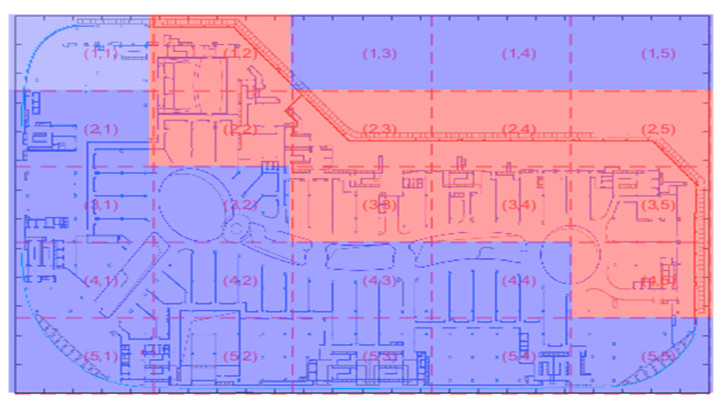
Risk heatmap.

**Figure 7 sensors-25-01481-f007:**
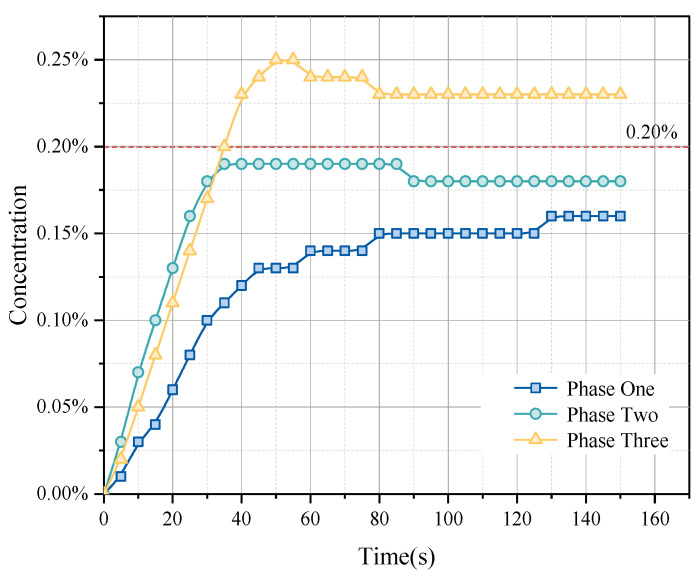
PDE optimization process diagram.

**Figure 8 sensors-25-01481-f008:**
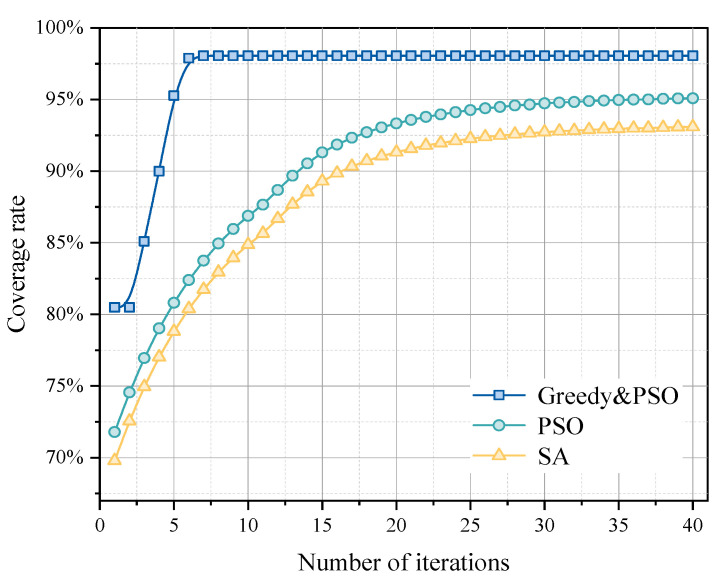
Comparison of optimization effects of different algorithms.

**Figure 9 sensors-25-01481-f009:**
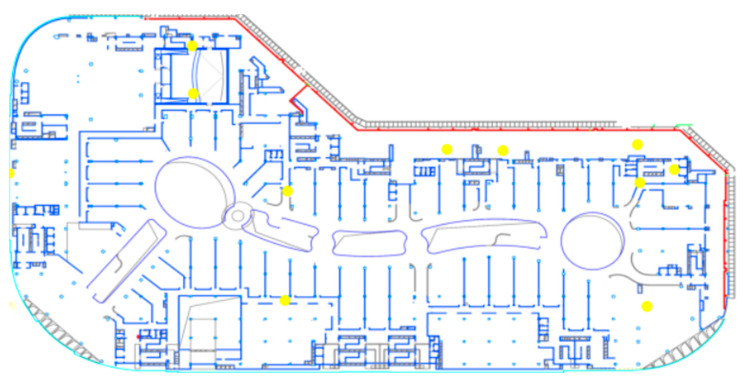
Sensor placement diagram.

**Figure 10 sensors-25-01481-f010:**
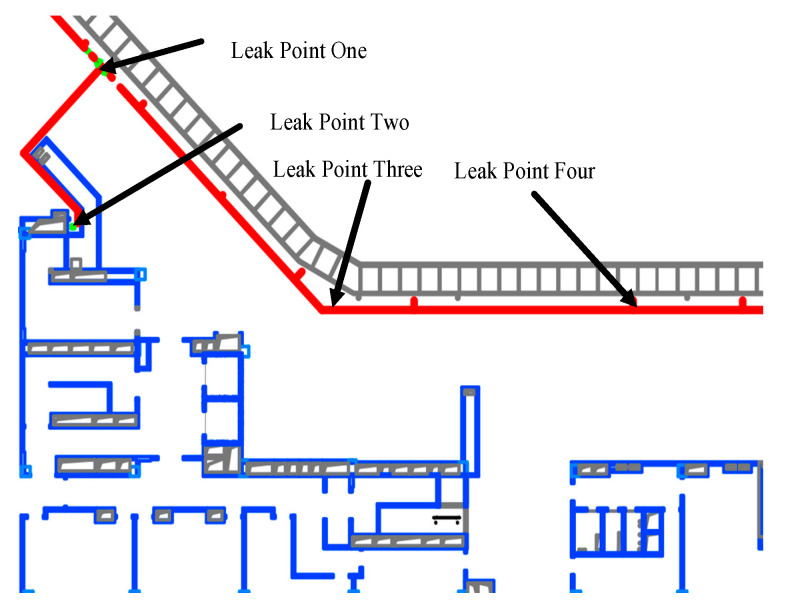
Schematic diagram of leakage location.

**Figure 11 sensors-25-01481-f011:**
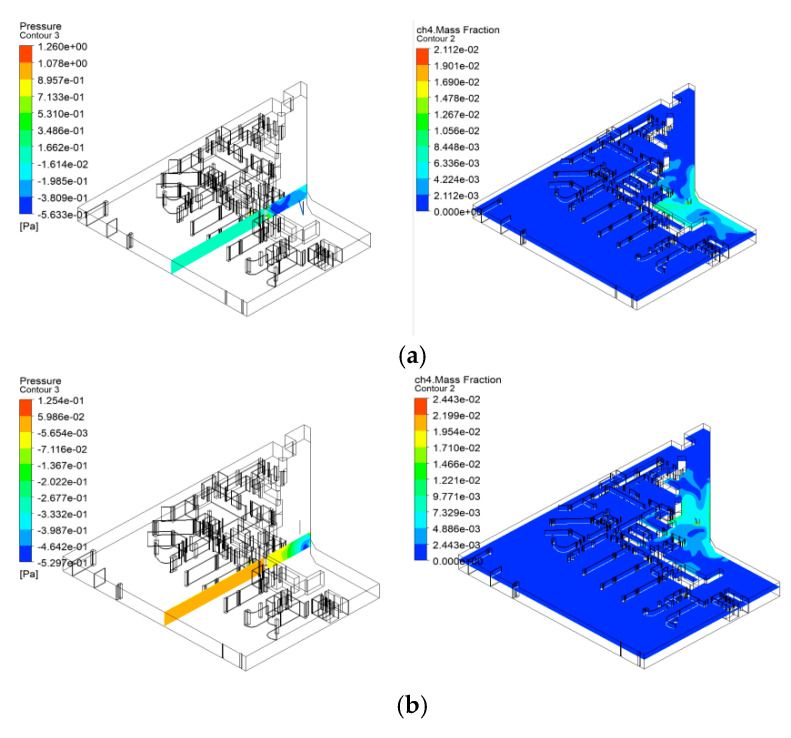
Simulation results diagram. (**a**) Methane composition graph for leak point one. (**b**) Methane composition graph for leak point two. (**c**) Methane composition graph for leak point three. (**d**) Methane composition graph for leak point four.

**Figure 12 sensors-25-01481-f012:**
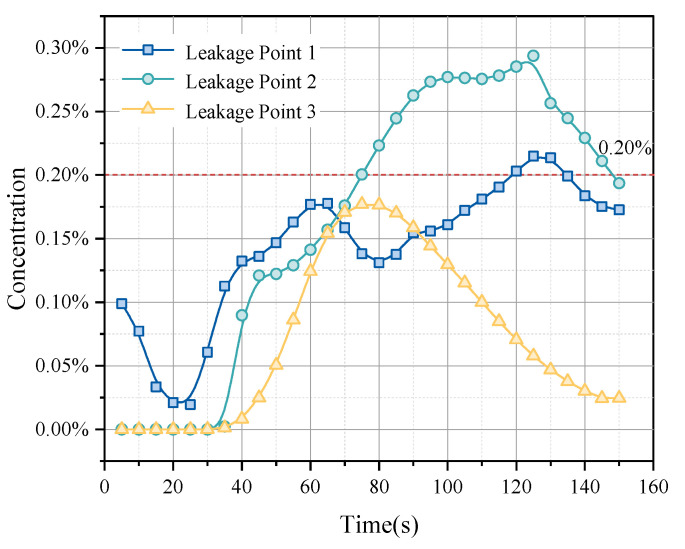
Methane component detection by sensors before optimization.

**Figure 13 sensors-25-01481-f013:**
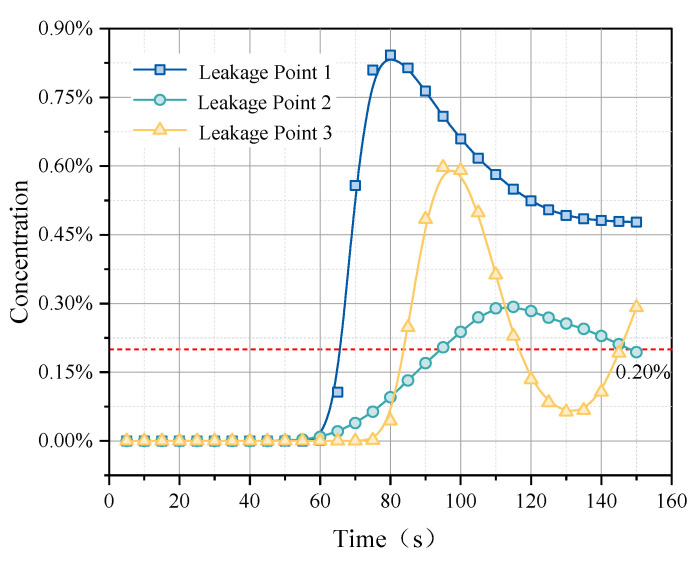
Methane component detection by sensors after optimization.

**Figure 14 sensors-25-01481-f014:**
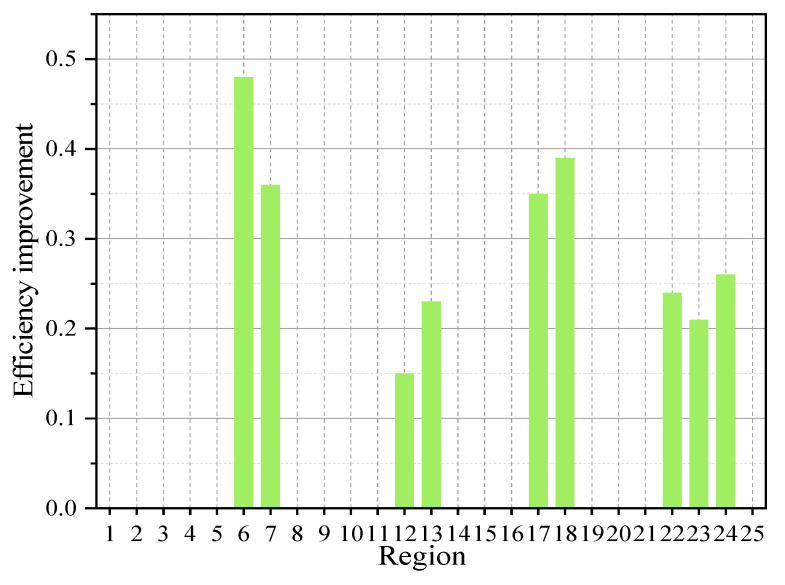
Optimization efficiency improvement diagram.

**Table 1 sensors-25-01481-t001:** Weighting criteria of a group of experts.

Professional Position	Duration of Experience	Educational Level	Age	Score
Advanced scholarly	≥30	PhD	-	5
Introductory academics	20–29	Master’s degree	≥50	4
Engineer	10–19	Bachelor’s degree	40–49	3
Technician	6–9	Associate degree	30–39	2
Worker	≤5	High school	<30	1

**Table 2 sensors-25-01481-t002:** Linguistic terms and fuzzy numbers.

Language Terminology	Triangular Fuzzy Number
Very high (VH)	(0.9, 0.95, 1)
High (H)	(0.75, 0.8, 0.85)
Near high (NH)	(0.55, 0.65, 0.75)
Medium (M)	(0.45, 0.5, 0.55)
Low (L)	(0.25, 0.35, 0.45)
Very low (VL)	(0.05, 0.15, 0.25)
Impossible (IM)	(0, 0.025, 0.05)

**Table 3 sensors-25-01481-t003:** Description of leakage factor nodes.

Node	Description	Node	Description	Node	Description
M1	Gas leak	X5	Valve status	X14	Operator experience
M2	Equipment status	X6	Sensor status	X15	Maintenance frequency
M3	Environmental conditions	X7	Sensor type	X16	Type of maintenance
M4	Operating procedures	X8	Sensor location	X17	Frequency of external inspections
M5	Maintenance activities	X9	Temperature conditions	X18	Compliance with operating procedures
X1	Pipeline status	X10	Humidity conditions	X19	Emergency plan
X2	Pipeline material	X11	Ventilation condition	X20	Training frequency
X3	Pipe joint	X12	Ventilation system	X21	Maintenance type
X4	Pipeline pressure	X13	Space volume	X22	Frequency of external inspections

**Table 4 sensors-25-01481-t004:** FPr, PT, and MT of root node.

Node	Condition	FPr %	PT %	MT %
X2	Steel	0.1	3.3	0.8
	Copper	0.1	2.3	1.2
	PVC	0.2	5.5	0.5
	Cast iron	0.1	4.3	0.6
X3	Well-sealed	0.1	2.3	0.9
	Loose	0.2	7.5	0.3
	Aging	0.3	1.1	0.2
X4	Low pressure	0.1	3.5	1
	Medium pressure	0.1	5.3	0.6
	High pressure	0.2	8.5	0.4
X5	New	0.1	1.8	1
	Normal	0.1	2.2	0.8
	Loose	0.2	6.3	0.4
	Blocked	0.2	23.5	0.3
	Corroded	0.2	1.1	0.2
X6	Normal	0.1	1.2	0.9
	Slight calibration deviation	0.1	3.5	0.6
	Severe calibration deviation	0.2	6.5	0.3
	Partial failure	0.2	9.3	0.2
	Complete failure	0.3	12.3	0.1
X9	Normal	0.1	1.5	1
	High temperature	0.1	4.5	0.4
	Low temperature	0.1	4.5	0.4
	Extreme high temperature	0.2	40.5	0.2
	Extreme low temperature	0.2	45.3	0.1
X10	Normal	0.1	1.8	1
	Humid	0.1	3.8	0.6
	Dry	0.1	3.8	0.6
	Extremely humid	0.2	6.8	0.3
	Extremely dry	0.2	6.4	0.3
X12	Good	0.1	1.8	1
	Moderate	0.1	3.2	0.6
	Poor	0.2	8.1	0.2
X14	Very abundant	0	1.1	1.2
	Experienced	0.1	2.5	1
	Average	0.1	4.3	0.8
	Less experienced	0.2	6.7	0.6
	Beginner	0.2	8.5	0.4
X16	Regular	0.1	2.5	0.8
	Occasional	0.2	5.3	0.4
	No maintenance	0.3	1.2	0.2
X18	Full compliance	0.1	2.1	1.1
	Partial compliance	0.1	5.6	0.8
	Non-compliance	0.2	10.1	0.6
X19	Established	0.2	1.1	0.7
	Not established	0.2	22.3	0.5
X20	Regular	0.1	5	0.7
X21	Preventive maintenance	0.1	6.3	0.4
	Corrective maintenance	0.1	20.1	0.35
	Emergency repair	0.2	31.3	0.2
X22	Monthly	0.1	1.1	0.5
	Quarterly	0.1	5	0.4
	Annually	0.2	35.3	0.1

**Table 5 sensors-25-01481-t005:** Relationship between risk levels and number of sensors.

R	Risk Level	Number of Sensor Placements
75 ≤ S < 100	Low risk	0
50 ≤ S < 75	Moderate risk	1
25 ≤ S < 50	Higher risk	2
0 ≤ S < 25	High risk	3

## Data Availability

Authors declare that no data has been published.

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
