# Peer review of "Integrated Bayesian and Particle Swarm Approaches for Enhanced Gas Leak Detection in Complex Commercial Structures"

_sensors, 2025, doi:10.3390/s25051481_

Round 1
Reviewer 1 Report
Comments and Suggestions for Authors
The paper addresses an important application of gas sensors regarding a gas leak detection in large spatial indoor spots. The suggested approach sounds interesting and the results match the scope of Sensors journal. However, prior the publication authors have to update the text to clarify several points as follows.
1. Authors have to clarify (Section 2) the type, producer and level of measured gas concentrations of the sensors applied in the study. They deliver some temporal characteristics in Figs. 12,13 which should be discussed accounting for a real response time of the sensors (p. 22). Possibly, they could provide a sensor signal vs time dependence (say, in Supplemental) for clarity.
2. In Section 2.1.1, authors introduce a group of “domain experts” which is absolutely unclear what for. Authors are advised to follow a linear logic and clarify the role of experts in teaching the network. This teaching has never been discussed in Section 3.
3. Authors have to check formulas provided on the subject of validity; see, for instance (10).
4. The data present in Figure 3 do not correspond to a description given in line 376. Authors are advised to plot inertia weight values.
5. Authors do not provide properly the data on which gas(es) is detected. At line 166, they note that the employed catalytic gas sensors are selective to ensure the detection of combustible analytes. However, this is not correct. As well documented in literature, the gas sensors, and particularly catalytic ones, are not so selective and can respond to numerous gases; see, for instance, Sensors and Actuators B, 306 (2020), 127615, DOI: 10.1016/j.snb.2019.127615. To enhance the selectivity, multisensor arrays are powerfully employed; see, for recent reviews, Measurement, 199 (2022), 111458, DOI: 10.1016/j.measurement.2022.111458; Sensors, 24 (2024), 4806, DOI: 10.3390/s24154806. Authors have to account for a selectivity issue in the state-of-art of the field discussed in Introduction.
6. As the major result, authors note (line 675 and Conclusion, line 692) enhancing the efficiency of gas detection by up to 47 %. However, it is unclear what is the “traditional layout”. This issue should be properly clarified and the methodic to estimate the efficiency should be present.
7. Figs. 12, 13 are unclear: is Y axis the sensor signal converted to a gas concentration? If so, it’s better to plot the sensor signal itself. The same relates to Fig. 7.
8. Fig. 14 is unclear: what is “Efficiency improvement”? Does it relate to Fig. 5? These data require a thorough discussion to let readers to understand.
9. Authors note (line 678) an efficiency of Computational Fluid Dynamics. To prove that they have to compare the data received without this technique.
Reviewer 2 Report
Comments and Suggestions for Authors
Thanks to the authors for a good manuscript. I read it with pleasure. A lot of research work was done and a large volume of manuscript was presented - 25 pages.
Optimization of the location of sensors (thermocatalytic) for detecting gas leaks is usually a simpler task, since it is tied to the infrastructure (gas pipe line). The authors provided a comprehensive optimization model for this problem. This is a good example for more complex research. For example more complicated task is the optimal location of gas fire detectors.
Similar simulations and full-scale experiments are carried out, for example, in aircraft design - there the issue of optimization consists not only of faster gas detection detection, but also of eliminating false positives. But a passenger aircraft is a more dynamic system (compared with buildings) and its model contains more Nodes.
In industrial buildings, the principle of "protecting the area" usually prevails, i.e. a fixed number of detectors per area of ​​the room (in my practice, optimization was performed only by the height of the location due to the different density of the detected gases relative to the ambient air). This approach is associated with insurance companies and the regulatory state framework. The implementation of optimized models of gas detector placement in a building is (in my practice) usually agreed with state emergency services and is an addition to the existing detection system, but not a replacement for one!
Good luck to the authors in their future research! I recommend publishing the manuscript in the present form.
Author Response
Dear Reviewer,
Thank you very much for your kind feedback and encouragement. We are honored to receive your recognition, and we are glad that you consider our sensor optimization model a valuable example for more complex research.
Regarding your comment on the principle of "protecting the area" commonly applied in industrial buildings, we will pay more attention to this issue in our future research, particularly focusing on how the optimized model can be practically implemented and effectively integrated with existing systems. We recognize that the optimized model should complement, not replace, the current detection system, and we will explore this aspect further in our future work.
Furthermore, your comparison with optimization issues in aircraft design has provided us with a valuable perspective. In our future research, we will consider how to apply optimization methods in various dynamic systems to further enhance the efficiency of detection systems.
Once again, thank you for your valuable comments. We will continue to consider and incorporate your suggestions in our future research. Best wishes for your continued success!
Sincerely,
Yuan Xiaobing
Reviewer 3 Report
Comments and Suggestions for Authors
Zhewen Sui et al. work "Integrated Bayesian and Particle Swarm Approaches for Enhanced Gas Leak Detection in Complex Commercial Structures" presents a method for placing sensors. This method uses a particle swarm optimization algorithm to determine the best number and position of sensors. The idea of the work is relevant to optimise the implementation of sensors in industry to achieve maximum safety in the presence of gas leaks. Before the manuscript is published, I suggest revising equations 1-17 and adding references if they were used based on other sources. It would also be useful to compare this study with other similar ones.
Author Response
Dear Reviewer,
Thank you for your valuable suggestions. In response to your feedback, we have revised Equation (17) in the manuscript to improve clarity and accuracy. Additionally, we have included the relevant reference to support this equation, as per your suggestion. The updated citation is as follows:
Kockarts, G. (2002). Transport phenomena. Journal De Physique Iv, 12(PR10), 235-252. https://doi.org/10.1051/jp4:20020462.
Furthermore, we have also expanded the discussion to include comparisons with other similar studies, which we believe will enhance the context and relevance of our work.
Thank you again for your helpful feedback. We hope that the revisions meet your expectations.
Round 2
Reviewer 1 Report
Comments and Suggestions for Authors
Authors have properly revised the paper regarding issues of concern.